# Rigid head-neck responses to unpredictable perturbations in patients with long standing neck pain does not change with treatment

**Ann-Katrin Stensdotter**[1]*, **Øyvind Stavdahl**[2], **Ottar Vasseljen**[3], **Ingebrigt Meisingset**[3]

**1** Faculty of Medicine and Health Sciences, Dept. of Neuromedicine and Movement Science Norwegian University of Science and Technology, NTNU, Trondheim, Norway, **2** Department of Engineering Cybernetics, Faculty of Information Technology and Electrical Engineering, The Norwegian University of Science and Technology, NTNU, Trondheim, Norway, **3** Department of Public Health and Nursing, Science, Faculty of Medicine and Health Sciences, The Norwegian University of Science and Technology, NTNU, Trondheim, Norway

* ann-katrin.stensdotter@ntnu.no

**Data Availability Statement:** The dataset for the current study is available at https://dataverse.no/dataset.xhtml?persistentId=doi:10.18710/AQCGJX.

## Abstract

In a previous study we have shown that patients with long standing non-specific neck-pain display more rigid neck movement behavior than controls in response to unpredictable perturbations. In the present study we investigated head/neck motor control in patients with neck-pain during a course of physiotherapy intervention and the associations with pain, neck disability and kinesiophobia. In this longitudinal observational study, 72 patients with non-specific neck-pain were exposed to unpredictable horizontal rotations by means of an actuated chair in three conditions; with a visual reference, and without vision with and without a cognitive task before first consultation with physiotherapist, after 2 weeks and 2 months of intervention. The neck movements were analyzed in the frequency domain to cover voluntarily and reflex controlled responses. Questionnaires encompassed Neck Disability Index, Tampa Scale of Kinesiophobia, and the Numerical Rating Scale for current pain. The results showed that the response pattern for the amplitudes of movement between head and trunk across frequencies did not change over time, whereas some changes in timing were found for some frequencies. Pain, neck disability, and kinesiophobia improved after intervention, but were not significantly associated with neck movement responses to perturbations across time or condition. Although physiotherapy intervention improved self-reported function, the rigid responses to unpredictable perturbations remained unchanged. This indicates altered function in reflex mediated control mechanisms, i.e., the vestibulocollic and the cervicocollic reflex systems that control the head in space and on the trunk. Future research should further investigate pain related changes in reflex systems and whether alterations in these systems are modifiable.

## Introduction

A comprehensive cross sectional study on long standing non-specific neck-pain has demonstrated that movement quality in patients was characterized by rigidity [1]. These findings

**Funding:** The Norwegian Fund for Post-Graduate Training in Physiotherapy through the FYSIOPRIM project. The funders had no role in study design, data collection and analysis, decision to publish, or preparation of the manuscript.

**Competing interests:** The authors have declared that no competing interests exist.

were corroborated by results in several earlier studies, showing reduced freedom of movement [2] as well as jerky and irregular cervical movements [3]. Kinetic measurements have shown deficits in direction specific force production, and neuromuscular recordings indicate increased muscle co-activation [4], delayed onset, and reduced activity in neck muscles [5]. Some studies have shown an association between altered neck kinematics and motor control and clinical symptoms such as pain, disability and kinesiophobia [6], while others have found only weak or no associations [1, 7].

Studies have in general assessed voluntary neck movements [3] and tasks, such as tracing an outlined figure [8], tracking an unpredictably moving target [9], or more commonly, cervical joint position errors [10]. Even though such tasks in general have shown reasonably good test-retest reproducibility [11], individual strategies will affect movement variability between as well as within subjects. Performance in voluntary tasks, even as simple as maximum voluntary isometric contraction, are subjected to practice and task specific learning effects that may explain improvement in performance [12]. Tests of motor control commonly use outcomes for movements precision in the time domain, as exemplified above, which provide information about performance but are hard to interpret in terms of underlying mechanisms. For the latter, measurements in the frequency domain are often used to analyze the responsiveness of a control system at various frequencies [13]. Furthermore, to reduce the impact of individual voluntary strategies as well as learning effects, protocols based on random and unpredictable perturbations are needed and has shown reasonably good reliability [14].

With such a protocol we have shown that patients with long standing non-specific neck-pain display rigid movement behavior compared to controls without neck pain when attempting to keep the head stationary in space [15]. At low frequencies <1Hz, head position in space can be voluntarily controlled [16–18], although reflex control is still active also at low frequency perturbations [19]. It has been debated whether or to what degree voluntary control can override these reflex responses [20]. At frequencies below 1 Hz, patients kept the head less steady in space but more steady relative to the trunk compared to asymptomatic controls [15]. At higher frequencies, reflexes stabilize the head on the trunk [16, 17]. In order to keep the head stationary in space, the head needs to counter-rotate relative to the trunk with the same amplitude and timing, which requires freedom of movement. Stiffer movement behavior has been, as mentioned above, found in several studies and some effects of therapeutic intervention on different motor control parameters has been demonstrated [11, 21]. No intervention studies seem so far to have tapped into the frequency domain and responses to unpredictable perturbations.

Our previous study, showing rigid responses to unpredictable perturbations in patients with neck-pain [15] is the departure point for the present study where we investigate head/neck motor control in the same cohort during a course of physiotherapy intervention. We hypothesized that rigidity of movement would be reduced, and the ability to keep the head stationary in space when exposed to unpredictable perturbations would improve at frequencies lower than 1 Hz. It was further hypothesized that findings of reduced pain, neck disability [7] and kinesiophobia after intervention would be associated with improved control of the head in space.

## Method

The present longitudinal observational study was a part of a cross-sectional study [1] and a prospective cohort study [7] on neck-pain where we reused patients and data for current pain and neck disability. The study was approved by the Regional Ethics Committee (2011/2522/REK) and conducted in agreement with the Helsinki declaration. Participants signed an informed consent before entering the study.

## Participants and setting

Patients from community (n = 61) and hospital (n = 21) physiotherapy clinics participated in this study in the period January 2013 to August 2014. Inclusion criteria were men and women with non-specific neck pain, age from 18 to 67 years old, pain duration >2 weeks, and average pain intensity at the day of testing $\geq$ 3 on a numerical rating scale (NRS 0–10, no to worst pain). Exclusion criteria were positive Spurling's test for neurological radiating arm pain, reduced and uncorrected vision or diagnosed vestibular deficits, history of neck trauma, orthopedic condition (e.g. previous neck surgery), or neurological conditions. The exclusion criteria were used to avoid influence on head/neck motor control from other medical conditions. Of a total of 145 invited patients, 72 of the 81 who were found eligible (characteristics described in Table 1), completed the test at baseline. Number of patients tested at 2 weeks and 2 months were 51 and 57, respectively. Twelve patients were tested only at baseline, and of these three patients dropped out due to illness, four without reason, and five did not respond. Patients received usual care physiotherapy and duration and number of treatments were at the discretion of the physiotherapists. Intervention was individualized as seen fit by the treating physiotherapist, and consisted of a wide range of modalities (percentage of patients who received the specific modalities in parentheses): individually supervised exercises (52%), massage (43%), mobilization/manipulation (45%), advice and information (27%), dry needling (23%), cognitive therapy (10%), and other modalities reported by less than 10% of the physiotherapists (exercises in group, prescribed home exercises, electrotherapy and shock wave therapy). Written and informed consent was obtained from all participants and the study was conducted in accordance to the Helsinki Declaration.

## Data acquisition

Self-administered questionnaires for issues regarding neck-pain and test of motor control of head/neck were completed before first physiotherapy consultation, at two weeks and after two months of physiotherapy treatment. Questionnaires encompassed Neck Disability Index (NDI; 0–100) [22], Tampa Scale of Kinesiophobia (TSK; 13–52) [23], and the NRS, 0–10, for current pain. Higher scores indicate worse symptoms.

Motor control for head/neck was investigated by asking patients to keep the head steady in space seated on a chair while the body was exposed to unpredictable random rotations of the chair in the horizontal plane (Fig 1). Three different conditions, each of 200 s duration, were administered in the following order; with vision (VS), without vision (NV), and without vision combined with a mental task counting backwards from 500 in steps of seven (MA). VS was designed to target voluntary control with visual guidance, NV without visual guidance relaying

**Table 1. Patients characteristics at baseline.** Values are reported as mean (SD), unless otherwise stated.

| Item | Baseline n = 72 |
|---|---|
| Females, n (%) | 51 (71) |
| Age (years) | 44.0 (12.8) |
| Height (m) | 1.70 (0.09) |
| Weight (kg) | 71.8 (14.4) |
| Body mass index (kg/m$^2$) | 24.5 (4.7) |
| Pain duration, n (%) | |
| < 3 months | 1. (10) |
| 3–12 Months | 22 (31) |
| >1 year | 43 (60) |

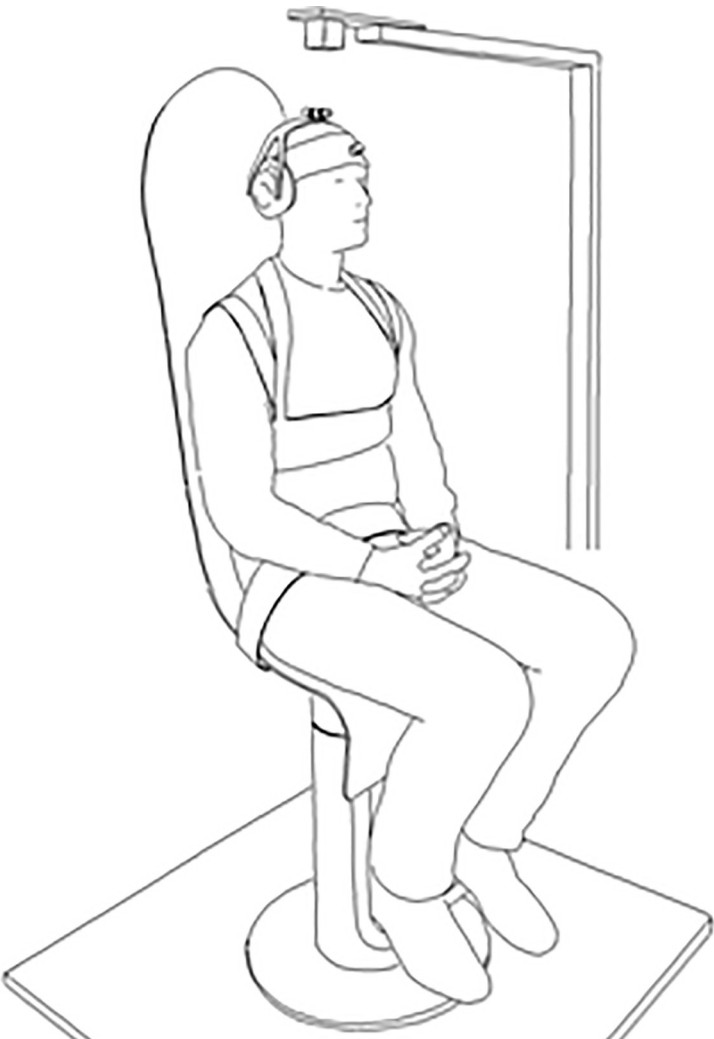

**Fig 1. An instrumented participant strapped to the actuated chair.** Sensors were placed on the chair, on the back of the subject at the level of the 2nd thoracic vertebrae, and on the forehead. The electromagnetic transmitter was placed ~ 20 cm above the head of the subject. Earmuffs have holes leaving the ears uncovered and hearing intact. Supporting information for the set up can be found in Stensdotter et al. [16].

only on proprioception, and MA to divert attention from conscious control of head position to investigate the contribution of reflex control. For visual reference, a laser pointer mounted in a rigid fixture on the head was aimed toward a vertical line on a white surface 1.6 m in front of the subject. A 5 cm intersecting horizontal line guided the projected laser beam keeping it aligned in the horizontal plane to retain the head tilt stable in neutral position preventing movements in the sagittal plane.

Sinusoidal rotational velocities around the vertical axis, coinciding approximately with the axis of the cervical spine, were induced to the trunk by means of an actuated chair. The patient was seated firmly strapped to the back and seat of the chair (Fig 1). Only the head was allowed free movement. The correspondence between frequencies induced by the chair and responses of the trunk has been validated for the experimental set up [16]. Movements were registered at 240 Hz with an electromagnetic motion tracking system (Liberty, Polhemus, Colchester, VT, USA, Fig 1).

The sum-of-sines excitation signal consisted of ten superimposed harmonic components chosen as prime multiples of a fundamental base frequency of 0.005 Hz over a fundamental period of $T = 200$ $s$. The prime numbers were chosen from the set $H$, where

$$H = \{37, 49, 71, 101, 143, 211, 295, 419, 589, 823\} \tag{1}$$

The same waveform was used for all conditions and all participants and provided pseudo-random perturbations in a pattern without repetitions preventing anticipatory preparation in the subjects (0.185 to 4.117 Hz). Chair velocity amplitudes were decreased as frequency increased: 20°/s from 0.185 Hz to 0.355 Hz, 19°/s from 0.505 Hz to 1.055 Hz, 16°/s from 1.475 Hz to 2.095 Hz, 15°/s at 2.945 Hz, and 13°/s at 4.115 Hz. The maximum rotational excursion occurred at the lowest frequency and was approximately ±17°. The formula for the sum-of-sines angular velocity excitation signal is described in Appendix 1 in S1 File. For a closer description, see Stensdotter et al 2016 [16].

## Data analysis

Information about the patients' motor responses to the perturbation was extracted with spectral analysis and the motion of the head and trunk would be a sum-of-sines in the excitation frequencies. Linearity was validated with spectral analyses of the head-room and trunk-room angles, showing satisfactory signal-to-noise ratio [16].

The system's frequency response was modelled as a complex transfer function, with the trunk angle taken as the excitation and the head angle taken as the response, both angles measured with respect to the room. Equations describing these transfer functions are found in Appendix 2 in S1 File). The term "spatial compensation" denotes the *gain* of this transfer function, i.e., the relative angular amplitude of the head and trunk in space. The phase angle of this transfer function relates to the relative *timing* of the head and trunk movements. Theoretically, perfect compensation for the head in response to the perturbations would be represented by a gain of zero, i.e., the head is kept stationary in space and thus has no angular amplitude relative to the room at the excitation frequency in question. (Keeping the head steady in space is achieved by head rotations relative to the trunk of the same amplitude and timing as, but in the opposite direction of, that of the trunk rotations relative to the room). Gain = 1 (identical motion amplitude in head and trunk) indicates that the head moves in space with the same angular excursion as the trunk, while gain > 1 indicates that the head moves more than the trunk relative to space. Perfect timing in response to perturbations would be shown as a phase angle of 0°; positive values denote phase lead and negative values indicate phase lag of the head in relation to the trunk. Resulting transfer functions are presented as Bode plots with gain and phase angle shown for the 10 excitation frequencies (Fig 2).

## Statistical analysis

The statistics were generated with STATA/IC 15.1. Normal distribution was confirmed for each separate frequency in each condition with histograms and Q-Q plots. Hierarchical linear mixed models, using the *mixed* command in STATA, were used to analyze if head/neck motor control changed from baseline to 2 weeks and 2 months (session 1 to 3, respectively), in the three different conditions. We assessed the overall effect of session for all frequencies (n = 10) in each condition (n = 3) for both phase and gain, that is whether phase and gain overall were significantly changed from session 1 to 2 and 3. The patient and the sessions (n = 3) were included as random effects. Different control systems are dominant in different frequency ranges, and patients may theoretically exhibit changes from session 1 to session 2 and 3 for frequencies related to voluntarily control systems, while the higher frequencies related to reflexive control may show no changes, and vice versa. The latter was investigated by including an

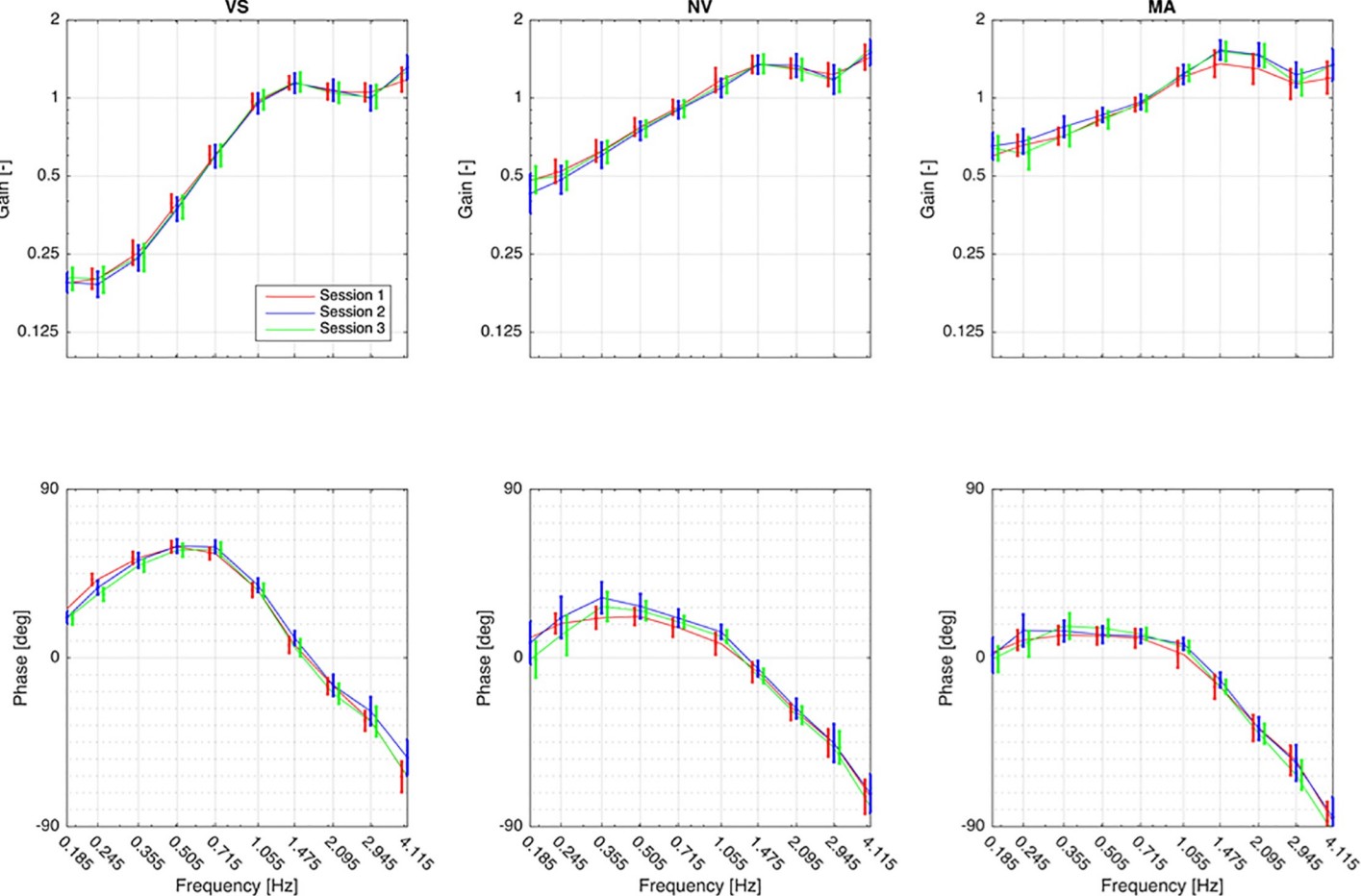

**Fig 2. Bode diagrams of transfer functions for the three conditions with vision (VS), without vision (NV), and without vision with a cognitive task (MA), respectively.** Mean and 95% CI. Plotted values are not adjusted for age and gender. Red line: before consultation with physiotherapist. Blue line: after two weeks of intervention. Green line: after two months of intervention. Gain = 0 (not attainable), the head is stationary in space; Gain = 1, the head is stationary on the trunk; Gain > 1, the head moves more than the trunk in space. Phase angle, positive values = phase lead, the head moves before the perturbations to the trunk; phase angle = 0°, perfect timing for compensation to perturbations; phase angle, negative values = phase lag, the head moves after the perturbation to the trunk.

interaction term between session and frequencies. To investigate if changes in pain intensity, disability, and kinesiophobia were associated with changes in head/neck motor control over time, we included these variables as covariates in the analyses of gain and phase for all conditions. Age, gender and body mass index were considered as confounders and therefore added as covariates in all models. A sample size calculation was not performed for the current study. The number of participants encompasses those included in previous publications by Meisingset and Stensdotter and widely exceeds the number in previous studies using this protocol [17]. The significance level was set to p<0.05.

## Results

Demographical and baseline clinical characteristics of the patients with neck-pain are provided in Table 1 and Table 2.

### Primary outcome variables

Fig 2 shows gain and phase for the three different conditions over time. Table 3 shows the results from the linear mixed models analyses for the different conditions.

**Table 2.  Clinical characteristics of patients with neck pain during a course of physiotherapy treatment.**  Values are reported as mean (SD).

|  | Baseline (n = 72) | 2 weeks (n = 51) | 2 months (n = 57) | LMM (p-value) |
|---|---|---|---|---|
| Current neck pain intensity (NRS; 0–10) | 4.7 (1.5) | 3.8 (1.7) | 3.1 (2.2) | <0.001 |
| Neck disability (NDI; 0–100) | 31.8 (12.1) | 23.0 (11.2) | 19.7 (12.5) | <0.001 |
| Kinesiophobia (TSK; 13–52) | 24.8 (4.2) | 22.7 (4.7) | 22.2 (4.8) | <0.001 |

NRS = numerical rating scale, higher score indicates more pain. NDI = Neck Disability Index, higher score indicates higher disability. TSK = Tampa Scale of Kinesiophobia, higher score indicates greater kinesiophobia. LMM = linear mixed model

**Gain.**   There was no significant overall change in gain from baseline to 2 weeks and 2 months for the conditions VS and NV (Table 3). In the MA condition, the gain was significantly higher at 2 weeks compared to baseline gain (β = 0.04; 95% CI = 0.01 to 0.06; p = 0.012), but the change was attenuated at 2 months (β = 0.02 (-0.004 to 0.05; p = 0.092). Effects on gain of the different perturbation frequencies did not change over the three sessions for any of the conditions, as documented by non-significant interaction terms between frequency and session.

**Phase angles.**   There was no significant overall change in phase angle from baseline to 2 weeks and 2 months for any condition (Table 3). However, the pattern of response for phase angles across frequencies and sessions showed some changes (Fig 2). Phase lag in the VS condition was significantly reduced at 2 weeks compared to baseline for frequency 2.945 Hz (β = 9.8; 95% CI = 1.3 to 18.4; p = 0.024) and frequency 4.115 Hz (β = 15.2; 95% CI = 6.7 to 23.7; p< 0.001). For the NV condition phase lead was significantly increased at 2 months compared to baseline for the frequencies 0.355 Hz (β = 17.5; 95% CI = 4.3 to 30.7; p = 0.009), 0.509 Hz (β = 14.9; 95% CI = 1.6 to 28.0; p = 0.027), 0.715 Hz (β = 15.0; 95% CI = 1.8 to 28.2; p = 0.026), and 1.055 Hz (β = 16.1; 95% CI = 2.9 to 29.3); p = 0.017).

## Secondary outcome variables

Pain intensity, neck disability and kinesiophobia were significantly reduced from baseline to 2 weeks and to 2 months (Table 2). Changes in these variables were not significantly associated with changes in gain and phase over time for the three conditions (estimates and p-values not reported), except for kinesiophobia in the VS condition where higher kinesiophobia was associated with smaller phase error across sessions (β = 0.4; 95% CI = 0.1 to 0.7, p = 0.021).

## Discussion

Our hypothesis that rigidity of movement would be reduced and the ability to keep the head stationary in space when exposed to unpredictable pseudorandom rotations would improve,

**Table 3.  Overall changes in head/neck motor control from baseline to 2 weeks and 2 months.**

|  | 2 weeks (n = 51) |  | 2 months (n = 57) |  |
|---|---|---|---|---|
|  | β (95% CI) | p | β (95% CI) | p |
| Gain |  |  |  |  |
| Visual (VS) | -0.002 (-0.03 to 0.02) | 0.90 | 0.003 (-0.02 to 0.03) | 0.79 |
| No vision (NV) | -0.01 (-0.04 to 0.01) | 0.31 | -0.006 (-0.03 to 0.02) | 0.65 |
| No vision-mental task (MA) | 0.04 (0.01 to 0.06) | 0.01 | 0.02 (-0.003 to 0.05) | 0.09 |
| Phase |  |  |  |  |
| Visual (VS) | 1.8 (-0.5 to 4.2) | 0.13 | -1.7 (-4.0 to 0.6) | 0.15 |
| No vision (NV) | 2.8 (-0.4 to 5.9) | 0.08 | -1.6 (-4.6 to 1.5) | 0.31 |
| No vision-mental task (MA) | 1.5 (-1.4 to 4.4) | 0.31 | -0.9 (-3.7 to 1.9) | 0.54 |

was not supported. The outcome after 2 weeks and after 2 months showed in general identical results for gain as well as for phase angles compared to baseline. Some scattered changes were however observed and the interpretation of the random nature of these is discussed below.

The relevance of increased gain in the MA condition at 2 weeks compared to baseline may be discussed because the size of this change was attenuated at 2 months and the change at 2 weeks was not associated with reduced pain, decreased disability, or reduced kinesiophobia. Furthermore, in a previous case-control study, gain was generally higher in the patient group and significantly so at 0.505 Hz [15]. A true increase in gain at the highest frequencies (> 3 Hz) would indicate reduced mechanical stiffness, meaning that the movement of the head would be greater than that of the trunk relative to the room. Likewise, relevance of changes in phase angle may be discussed. Although reduced phase lag above 2 Hz in the VS condition at 2 weeks and increased phase lead below 1.5 Hz in the NV condition at 2 months did resemble a pattern for healthy controls, our previous study has shown no significant differences for these outcomes between patients and controls [15]. In addition, the association found between kinesiophobia and reduced phase lag in VS appears illogical as it suggests that greater fear of movement associates with phase angles more similar to those of controls. Finally, increased phase lead in NV was not associated with kinesiophobia.

Phase lead observed below 1.5 Hz in the current study has been observed in several other studies, but there are no definite explanations for this as anticipation is not likely in these types of protocols [16, 17, 24]. It has however been suggested that phase lead may be explained by reflex induced dampening of the system dynamics [24].

Altogether, the main findings from this study suggest that head-neck control in response to unpredictable perturbations did not change after physiotherapy intervention, even at frequencies below 1 Hz where voluntary control is possible and where in our previous study we found differences between patients and healthy controls [15]. In contrast, in another study, the same cohort has shown improved accuracy of controlling the head-neck system in different *voluntarily* controlled tracing and tracking tasks [7]. Thus, within the same cohort, no transfer effect was found between improvement in voluntary tasks [7] and responses to unpredictable perturbations presented in the current study. We have not been able to find any treatment studies that compare to our protocol, that is, a comprehensible prospective protocol including both perturbation responses, voluntary control tasks, and self-reports on pain and function. With regard to voluntary tasks and self-reports, several studies have reached comparable results to Meisingset (2016), such as improved performance in direction specific muscle activation or reduced joint position error after intervention, additionally accompanied by reduced pain and disability [21, 25]. A meta-analysis on the effect of motor control exercises, specifically on exercises for the deep cervical flexors, reveals effect on motor control in addition to pain and disability [26]. Thus, there seem to be some evidence for associations between improved voluntary control and self-reported pain and function. Another study has shown associations between different tasks, demonstrating transfer effects from improved motor control in learning to balance a metal ball on flat surface strapped on top of the head to reduced postural sway and jerkiness of cervical rotation [27]. This result was explained by the authors as a possible effect on neck proprioception which is essential to postural control through the interlinkage of muscle spindle input and integration in the central nervous system [28]. In theory, an effect of exercise intervention to improve neck muscle proprioception may explain better performance of voluntary neck movement and the transfer to postural control, presumably by reduced activity in muscle spindle afferents and reduced gamma motor-neuron activity [29]. Although a reduction of postural sway was also found in our cohort (Meisingset et al 2016), our protocol differed in several important aspects; The intervention was individualized as seen fit by the treating physiotherapist, and consisted of a wide range of modalities [7], i.e., no task specific

training was included specifically aimed toward improving task performance in the test protocol.

Some studies have tried to explain changes in motor control related to pain conditions. There is some evidence that already a first episode of acute neck-pain may cause alterations in neuromuscular behavior [30]. Animal studies suggest that experimental pain renders the spinal cord resistant to plastic changes for learning [31], at least as long as the pain remains. Furthermore, a vicious circle is suggested where increased muscle tension produces metabolites activating gamma motor neurons, increasing activity in muscle spindle afferents and again increasing muscle stiffness [29], potentially explaining rigid motor responses in neck-pain Pain induced increase in gamma motor-neuron activity has been corroborated in studies of experimental muscle pain showing increased amplitude of the stretch reflex. Experimentally induced pain does however not support the notion of a vicious pain circle as it does not demonstrate a corresponding increase in the H-reflex amplitude, probably due to the short-term effect [32]. Nevertheless, a reflex mediated increase of muscle stiffness would restrict freedom of movement for the head and neck necessary to voluntarily compensate for perturbations to the trunk when attempting to keep the head steady in space. This reasoning is corroborated by other studies on neck-pain [33, 34], suggesting increased co-activation between agonist and antagonist muscles of the neck in response to unpredictable perturbations.

With regard to the above theory on pain induced spinal cord affection, hypersensitivity in several of the neck muscles [35] may explain the episodic course and recurrence in non-specific neck-pain [36] and presentation of altered motor behavior or deficits in motor control of the cervical spine [37]. Our results showing no changes in responses to unpredictable perturbations substantiate the suggestion of changes in reflex mediated motor control.

Limitations: Muscle activity was not monitored in this study to verify changes in muscle stiffness to support assumptions on up-regulated gamma-motor neuron activity [29]. Further studies need to include electromyography to assess associations between muscle activity and system dynamics.

In conclusion, the present results suggest that although physiotherapy intervention improves pain and voluntary movement control, responses to unpredictable perturbations remain generally unchanged. This indicates the presence of altered function in reflex mediated control mechanisms, i.e., the vestibulocollic and the cervicocollic reflex systems that control the head in space and the head on the trunk, respectively [38, 39]. Future research should further investigate changes in reflex systems and whether presumed alterations in these systems are modifiable.

## Supporting information

**S1 File.**
(DOCX)

## Acknowledgments

We kindly thank Dr. Ing. Morten Dinhoff Pedersen for assistance with the analyses and bode plots, and Prof. Turid Follestad for statistical advices.

## Author Contributions

**Conceptualization:** Ann-Katrin Stensdotter.

**Data curation:** Ann-Katrin Stensdotter, Øyvind Stavdahl.

**Formal analysis:** Ann-Katrin Stensdotter, Ingebrigt Meisingset.

**Funding acquisition:** Ottar Vasseljen.

**Investigation:** Ingebrigt Meisingset.

**Methodology:** Ann-Katrin Stensdotter, Øyvind Stavdahl.

**Project administration:** Ottar Vasseljen.

**Resources:** Øyvind Stavdahl, Ottar Vasseljen.

**Software:** Øyvind Stavdahl.

**Validation:** Øyvind Stavdahl.

**Writing – original draft:** Ann-Katrin Stensdotter.

**Writing – review & editing:** Øyvind Stavdahl, Ottar Vasseljen, Ingebrigt Meisingset.

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
