## [Editor Report · Decision Letter 0]

28 Apr 2020

PONE-D-20-06059

Rigid head-neck responses to unpredictable perturbations in patients with long standing neck pain does not change with treatment.

PLOS ONE

Dear Professor Stensdotter,

Thank you for submitting your manuscript to PLOS ONE. After careful consideration, we feel that it has merit but does not fully meet PLOS ONE’s publication criteria as it currently stands. Therefore, we invite you to submit a revised version of the manuscript that addresses the points raised during the review process.

We would appreciate receiving your revised manuscript by Jun 12 2020 11:59PM. To enhance the reproducibility of your results, we recommend that if applicable you deposit your laboratory protocols in protocols.io, where a protocol can be assigned its own identifier (DOI) such that it can be cited independently in the future. For instructions see: http://journals.plos.org/plosone/s/submission-guidelines#loc-laboratory-protocols

We look forward to receiving your revised manuscript.

Kind regards,

Deepak Kumar

Academic Editor

PLOS ONE

Additional Editor Comments:

study is up to mark but require some changes

1. elaborate inclusion and exclusion criteria

2. mention method of sample sample calculation

3. written consent of participants taken or not

4. age group is not mentioned

5. specify physiotherapy intervention
---

## [Author Response · Author response to Decision Letter 0]

28 May 2020

Dear Dr. Deepak Kumar,

We kindly thank you as the academic editor for valuable comments to improve our manuscript and the opportunity to submit a revised version of our manuscript “Rigid head-neck responses to unpredictable perturbations in patients with long standing neck pain does not change with treatment”. by Ingebrigt Meisingset, Ottar Vasseljen, Øyvind Stavdahl, and Ann-Katrin Stensdotter. We hope that our amendments are satisfactory for acceptance in PLOS ONE.

We have uploaded a revised manuscript with track changes labeled 'Revised Manuscript with Track Changes' and an unmarked revised version where all track changes are removed labeled “Manuscript”. 

Below you will find our response to each point raised by the academic editor.

1. elaborate inclusion and exclusion criteria

• We agree that several of the criteria we used were missing in the manuscript, and have added and further described those (lines 80-87 in marked version)

2. mention method of sample calculation

• There was no a priori sample size calculation for this study, as this study was part of an observational study of patients with neck pain investigating a comprehensive set of tests for neck motion and motor control. This information is already given in the section statistical analysis.

3. written consent of participants taken or not

• A written consent was signed by all participants before entering the study. This has been added in section participants and setting (line 99-101 in marked version)

4. age group is not mentioned

• The age range has been added in inclusion criteria (see also pt. 1 above)

5. specify physiotherapy intervention

• We referenced to our previous study for this information. We have now specified the physiotherapy intervention in the section participants and setting (lines 92-99 in the marked version).

• We have, as far as we can see, adhered to the stye as described in PLOS One formatting guidelines and have added line numbers as we noticed that we had forgotten.

If there are ethical or legal restrictions on sharing a de-identified data set, please explain them in detail (e.g., data contain potentially identifying or sensitive patient information) and who has imposed them (e.g., an ethics committee). Please also provide contact information for a data access committee, ethics committee, or other institutional body to which data requests may be sent.

• We have added a section in the manuscript with information of data sharing and contact information for request of data sharing.

• See point above

• We have added a caption for the Supporting Information file after the references and have made the in-text citations clearer.

---

## [Editor Report · Decision Letter 1]

5 Aug 2020

Rigid head-neck responses to unpredictable perturbations in patients with long standing neck pain does not change with treatment.

PONE-D-20-06059R1

Dear Dr. Stensdotter,

We’re pleased to inform you that your manuscript has been judged scientifically suitable for publication and will be formally accepted for publication once it meets all outstanding technical requirements.

Kind regards,

Bernadette Ann Murphy, PhD

Academic Editor

PLOS ONE
---

## [Editor Report · Acceptance letter]

14 Aug 2020

PONE-D-20-06059R1 

Rigid head-neck responses to unpredictable perturbations in patients with long standing neck pain does not change with treatment. 

Dear Dr. Stensdotter:

I'm pleased to inform you that your manuscript has been deemed suitable for publication in PLOS ONE. Congratulations! Your manuscript is now with our production department. 

Kind regards, 

on behalf of

Dr. Bernadette Ann Murphy 

Academic Editor

PLOS ONE